

# The pomegranate-derived peptide Pug-4 alleviates nontypeable *Haemophilus influenzae*-induced inflammation by suppressing NF-kB signaling and NLRP3 inflammasome activation

Pornpimon Jantaruk[1], Sittiruk Roytrakul[2], Anchalee Sistayanarain[1] and Duangkamol Kunthalert[1,3]

[1] Department of Microbiology and Parasitology, Faculty of Medical Science, Naresuan University, Phitsanulok, Thailand
[2] National Science and Technology Development Agency, Thailand Science Park, National Center for Genetic Engineering and Biotechnology, Pathumthani, Thailand
[3] Centre of Excellence in Medical Biotechnology, Faculty of Medical Science, Naresuan University, Phitsanulok, Thailand

Corresponding author
Duangkamol Kunthalert,
duangkamolk@nu.ac.th,
kunthalertd@yahoo.com

## ABSTRACT

The respiratory pathogen nontypeable *Haemophilus influenzae* (NTHi) is the most common cause of exacerbation of chronic obstructive pulmonary disease (COPD), of which an excessive inflammatory response is a hallmark. With the limited success of current medicines there is an urgent need for the development of novel therapeutics that are both safe and effective. In this study, we explored the regulatory potential of pomegranate-derived peptides Pug-1, Pug-2, Pug-3, and Pug-4 on NTHi-induced inflammation. Our results clearly showed that to varying degrees the Pug peptides inhibited NTHi-induced production of IL-1$\beta$, a pivotal cytokine in COPD, and showed that these effects were not related to cytotoxicity. Pug-4 peptide exhibited the most potent inhibitory activity. This was demonstrated in all studied cell types including murine (RAW264.7) and human (differentiated THP-1) macrophages as well as human lung epithelial cells (A549). Substantial reduction by Pug-4 of TNF-$\alpha$, NO and PGE$_2$ in NTHi-infected A549 cells was also observed. In addition, Pug-4 strongly inhibited the expression of nuclear-NF-$\kappa$B p65 protein and the NF-$\kappa$B target genes (determined by IL-1$\beta$, TNF-$\alpha$, iNOS and COX-2 mRNA expression) in NTHi-infected A549 cells. Pug-4 suppressed the expression of NLRP3 and pro-IL-1$\beta$ proteins and inhibited NTHi-mediated cleavage of caspase-1 and mature IL-1$\beta$. These results demonstrated that Pug-4 inhibited NTHi-induced inflammation through the NF-$\kappa$B signaling and NLRP3 inflammasome activation. Our findings herein highlight the significant anti-inflammatory activity of Pug-4, a newly identified peptide from pomegranate, against NTHi-induced inflammation. We therefore strongly suggest the potential of the Pug-4 peptide as an anti-inflammatory medicine candidate for treatment of NTHi-mediated inflammation.

# INTRODUCTION

Nontypeable *Haemophilus influenzae* (NTHi) is a Gram-negative, non-encapsulated bacterium that is typically a commensal of the human nasopharynx. NTHi can become an opportunistic pathogen causing a range of diseases including middle ear infections (otitis media), conjunctivitis, sinusitis, and community-acquired pneumonia (*Agrawal & Murphy, 2011*). NTHi is also a major cause of the exacerbation of chronic obstructive pulmonary disease (COPD) (*Sriram et al., 2018*; *Van Eldere et al., 2014*), a disease that is the main leading cause of human morbidities and mortalities worldwide (*Short et al., 2021*; *Su et al., 2018*). Although cigarette smoking is the main environmental factor to trigger COPD, bacterial infection is also a common phenomenon with approximately half of COPD cases attributed to NTHi (*Agrawal & Murphy, 2011*; *Barnes et al., 2015*). COPD is a major public health problem that negatively impacts a patient's quality of life and contributes to extensive healthcare and economic costs (*Halpin et al., 2017*; *Hurst et al., 2021*; *Iheanacho et al., 2020*).

COPD is an airway inflammatory disease characterized by long-term respiratory symptoms and airflow limitation (*Mirza et al., 2018*). The presence of bacteria, in particular NTHi, additionally increases the inflammation levels in the airways of COPD patients (*Saliu et al., 2021*; *Sethi et al., 2006*). The proinflammatory cytokine interleukin (IL)-1$\beta$ is primarily involved in the initiation and persistence of airway inflammation and has been found to be correlated with COPD (*Fu et al., 2015*; *Guo et al., 2022*; *Zou et al., 2017*). IL-1$\beta$ secretion is elevated in both stable and exacerbated cases of COPD (*Bafadhel et al., 2011*; *Pauwels et al., 2011*). A markedly positive association between serum IL-1$\beta$ and IL-17 levels in COPD patients with acute exacerbation has also been described (*Zou et al., 2017*). Unlike other cytokines, the synthesis and secretion of mature IL-1$\beta$ is under tight transcriptional and post-transcriptional control that includes the involvement of the nucleotide-binding oligomerization domain-like receptor containing pyrin domain 3 (NLRP3) inflammasome (*Kelley et al., 2019*). The NLRP3 inflammasome is a cytosolic protein complex that functions as a molecular platform for caspase-1-dependent proteolytic maturation and secretion of IL-1$\beta$ and IL-18 (*Akbal et al., 2022*). Accumulating evidence has demonstrated that NLRP3 inflammasome activation is involved in the airway inflammation observed in patients with COPD. NLRP3 inflammasome activation and its ensuing products are present in high levels in the bronchoalveolar lavage, sputum and lung tissues of COPD patients (*Colarusso et al., 2017*). Moreover, local airway NLRP3 inflammasome activation is positively correlated with acute exacerbations and lower airway microbial colonization in COPD patients (*Colarusso et al., 2017*; *Wang et al., 2018*). Importantly, NLRP3 inflammasome is upregulated during a NTHi infection in respiratory cells and tissues and associated with caspase-1-dependent IL-1$\beta$ release (*Rotta Detto Loria et al., 2013*). This indicates that the inflammasome is activated in bacterial-driven exacerbations in COPD. To date there has been a significantly increased interest in the NLRP3 inflammasome as a new era therapeutic target for regulating NLRP3-driven inflammation including that present in COPD exacerbations (*Chernikov et al., 2021*; *Leszczyńska, Jakubczyk & Górska, 2022*; *Swanson, Deng & Ting, 2019*; *Zhang et al., 2021*).

Inhaled corticosteroids and bronchodilators are common pharmaceuticals prescribed to manage the symptoms of COPD. Their clinical uses are, however, limited due to the undesirable side effects and high costs of these drugs (*Santos et al., 2016*). Of significant note, NTHi-induced steroid resistance in COPD patients has also been documented (*Cosío et al., 2015*; *Khalaf et al., 2017*). The adverse side effects and resistance to the current medicines present an urgent need for the development of novel, safe, and effective therapeutics.

Pomegranate (*Punica granatum* L.) is one of the oldest known edible fruits that is widely consumed and is well known for its medicinal properties. Pomegranate is a rich source of a variety of phytochemicals including phenolic acids, tannins, and flavonoids (*Vučić et al., 2019*). Research so far has shown that pomegranate possesses a variety of pharmacological benefits such as anti-oxidant, anti-microbial, anti-inflammatory, anti-cancer, anti-fibrotic, and anti-angiogenic properties as well as other benefits (*BenSaad et al., 2017*; *Jalali et al., 2021*; *Maphetu et al., 2022*; *Rakhshandeh et al., 2022*; *Sharma et al., 2022*). *In vitro* and *in vivo* studies have also revealed the potential role of pomegranate in treating lung-based diseases (*Shaikh & Bhandary, 2021*). Apart from phytochemicals, peptides and protein hydrolysates from pomegranate have been studied for their potential health benefits (*Guo, Wang & Ng, 2009*; *Kokilakanit et al., 2020*). Their biological activity, high specificity, low toxicity, and low production costs make the bioactive peptides of great interest in pharmaceutical research and development (*Fosgerau & Hoffmann, 2015*; *Wang et al., 2022*). In recent years, protein hydrolysates from dried pomegranate fruits were prepared, partially purified, and sequence identified (*Kokilakanit et al., 2020*). Subsequently, the peptides Pug-1, Pug-2, Pug-3 and Pug-4 in the highest antibacterial fraction were selected, synthesized, and tested for an anti-biofilm effect on *Streptococcus mutans* adhesion (*Kokilakanit et al., 2020*). The immunomodulatory activity of such peptides from pomegranate has not yet been reported. In this study, the anti-inflammatory potential of the pomegranate-derived peptides in NTHi-induced inflammation were investigated. The possible molecular mechanisms involved in its action were also explored. This study demonstrates that a pomegranate-derived peptide, Pug-4 is a promising therapeutic for treatment of NTHi-mediated inflammation.

## MATERIAL AND METHODS

### Peptides

The pomegranate-derived peptides Pug-1 (LLKLFFPFLETGE), Pug-2 (GAVGSVV), Pug-3 (LGTY) and Pug-4 (FPSFLVGR) with the purity > 90% were synthesized by ChinaPeptides Co., Ltd. (Shanghai, China) or GenScript (Piscataway, NJ, USA). All synthetic peptides were dissolved in ≥ 99.5% dimethyl sulfoxide (DMSO; Sigma, Saint-Quentin-Fallavier, France) and further diluted in fresh antibiotic-free culture medium to obtain the required working concentrations.

### Bacterial strain and culture condition

This study used nontypeable *Haemophilus influenzae* (NTHi) NU74, a clinical isolate previously identified by microbiological and biochemical procedures and serotyped by PCR (*Kunthalert & Kamklon, 2007*). Bacteria were grown on chocolate agar consisting of

a GC medium base (Difco™; Becton Dickinson, Franklin Lakes, NJ, USA), 2% isovitalex (Becton Dickinson, Franklin Lakes, NJ, USA), and 2% hemoglobin (Becton Dickinson, Franklin Lakes, NJ, USA) at 37 °C in 5% $CO_2$. Following overnight incubation, the bacteria were harvested and washed twice with sterile phosphate-buffered saline (PBS) pH 7.4. Bacterial suspensions were then prepared in an appropriate culture medium for subsequent experiments.

## Bacterial viability assay

The effect of Pug peptides on the viability of NTHi was determined by the minimum inhibitory concentration (MIC) and the minimum bactericidal concentration (MBC). Two-fold serial dilutions of Pug peptides were prepared in a Haemophilus test medium (HTM) in 96-well microtiter plates (Nunc, Roskilde, Denmark). An adjusted bacterial inoculum was then added to each well to achieve a final concentration of $5 \times 10^5$ CFU/mL. The final concentrations of peptides ranged from 0.05 µM to 100 µM. Bacterial culture without the test peptides served as an untreated control. The MIC was determined as the lowest peptide concentration that yielded no visible growth after incubation at 37 °C in a 5% $CO_2$ atmosphere for 24 h. To determine MBC, 10 µL was aspirated from wells where there was no visible growth in the MIC experiment and these were then plated onto chocolate agars. The plates were incubated at 37 °C under 5% $CO_2$ for 24 h and the MBC was determined as the lowest peptide concentration at which no bacterial growth was observed.

## Cell cultures

The murine macrophage cell line RAW264.7 (TIB-71), human monocytic THP-1 cells (TIB-202), and human lung epithelial cells A549 (CCL-185) were obtained from American Type Culture Collection (ATCC, Manassas, VA, USA). The base medium for RAW 264.7 was DMEM (HyClone, Utah, USA) while RPMI-1640 medium (HyClone) was used for THP-1 and A549 cells. To make the complete growth medium, 10% (v/v) heat-inactivated fetal bovine serum (FBS; Gibco, Waltham, MA, USA), 10 mM 4-(2-hydroxyethyl)-1-piperazineethanesulfonic acid (HEPES; HyClone™, Utah, USA), 2 mM L-glutamine (PAA Laboratories GmbH, Pasching, Austria), 100 U/mL penicillin, and 100 µg/mL streptomycin (PAA) were added to the respective base medium. A supplementation of 2-mercaptoethanol (Bio-Rad Laboratories, Hercules, CA, USA) with a final concentration of 0.05 mM was applied to the complete RPMI-1640 medium for growth of THP-1 cells. These cell lines were cultured at 37 °C in a humidified atmosphere of 5% $CO_2$.

The THP-1 monocytes ($1 \times 10^5$ cells/well) in 96-well plates (Nunc™) were differentiated into macrophage-like cells by 24 h-treatment with 50 ng/mL phorbol 12-myristate 13-acetate (PMA; Sigma-Aldrich, St. Louis, MO, USA). The cells were then washed once with PBS pH 7.4 to discard nonadherent cells and subsequently incubated in complete RPMI-1640 for 24 h at 37 °C in a 5% $CO_2$ humidified atmosphere. The differentiated THP-1 cells were washed three times with PBS pH 7.4, maintained in antibiotic- and serum-free RPMI-1640 medium, and used in subsequent experiments.

## Cell viability assay

A tetrazolium salt 3-(4,5-dimethylthiazol-2-yl)-2,5-diphenyltetrazolium bromide (MTT) reduction assay described previously (*Mosmann, 1983*) was used to determine cytotoxicity of the Pug peptides. Three cell lines were used in this assay. The cells at a density of 1 $\times$ $10^5$ cells/well in 96-well plates (Nunc™) were treated with the test peptides with the concentrations varying from 25 µM to 100 µM, or vehicle for 24 h at 37 °C in 5% $CO_2$. Cells without the test peptides served as an untreated control. After 24 h-incubation at 37 °C in 5% $CO_2$, 20 µL of 5 mg/mL MTT (Sigma, St. Louis, MO, USA) was added to each well and plates were incubated for another 3 h to allow reduction of MTT. After the supernatant was discarded, the insoluble formazan crystals were solubilized by adding 100 µL DMSO and gently shaking for 15 min. The optical density (OD) at 540 nm was then measured using a microplate reader (Rayto RT-2100C). The percentage of cell viability was determined according to the equation: (OD of treated cells/OD of untreated cells) $\times$ 100.

## Peptide treatment and NTHi infection protocol

RAW 264.7, differentiated THP-1, or A549 cells at a density of 1 $\times$ $10^5$ cells/well in 96-well plates (Nunc™) or 2 $\times$ $10^6$ cells/well in 6-well plates (Nunc™) maintained in antibiotic- and serum-free medium were pre-treated with the test peptides at concentrations ranging from 2.5 µM to 100 µM for 1 h at 37 °C in 5% $CO_2$. Cells were subsequently infected with NTHi at a multiplicity of infection (MOI) of 10 for 6 h (differentiated THP-1), 9 h (A549), or 24 h (RAW264.7). Culture supernatants were collected by centrifugation at 1,500 rpm for 10 min at 4 °C to quantify the inflammatory responses.

## Cytokine quantification

The concentrations of IL-1$\beta$ and TNF-$\alpha$ in the culture supernatant were quantified by a sandwich enzyme-linked immunosorbent assay (ELISA) using ELISA MAX™Deluxe Set (BioLegend, San Diego, CA, USA) according to the manufacturer's instructions.

## Measurements of nitric oxide and PGE$_2$

The levels of nitrite (a stable breakdown product of nitric oxide (NO)) in the culture supernatant were measured using the Griess reagent system (Promega, Madison, WI, USA) according to the manufacturer's instructions. The PGE$_2$ levels in the culture supernatant were determined by using an enzyme-immunoassay (R&D Systems, Minneapolis, MN, USA) following the manufacturer's protocols.

## Quantitative real-time polymerase chain reaction (qPCR) analysis

A549 cells at 2 $\times$ $10^6$ cells/well in 6-well plates (Nunc™) were pre-treated with various concentrations of the Pug-4 peptide (25–100 µM) for 1 h at 37 °C in 5% $CO_2$ and infected with NTHi at a MOI of 10. Following a 9-h incubation, the total RNA was isolated from A549 cells using TRIzol reagent (Invitrogen, Waltham, MA, USA) and the total RNA quantity was evaluated using a Nanodrop spectrophotometer (NanoDrop Technologies, USA). Thereafter, 2 µg of extracted RNA was reverse transcribed into cDNA using the RevertAid First Strand cDNA Synthesis kit (Thermo Fisher Scientific) as recommended by the manufacturer. The qPCR was performed using the SYBR GreenStar qPCR Master Mix

**Table 1  Primers used for qPCR analysis in this study.**

| Target gene | Primer sequence (5′–3′) | Tm (°C) | References |
|---|---|---|---|
| IL-1β | Forward 5′-CAG CCA ATC TTC ATT GCT CA- 3′ | 53.3 | *Ye et al. (2015)* |
| | Reverse 5′-TCG GAG ATT CGT AGC TGG AT- 3′ | 55.2 | |
| TNF-α | Forward 5′-CTC TTC TGC CTG CTG CAC TT- 3′ | 57.7 | *Panepucci et al. (2007)* |
| | Reverse 5′-GCC AGA GGG CTG ATT AGA GA- 3′ | 55.6 | |
| iNOS | Forward 5′-TGG ATG CAA CCC CAT TGT C- 3′ | 55.7 | *Törnblom et al. (2005)* |
| | Reverse 5′-CCC GCT GCC CCA GTT T- 3′ | 68.8 | |
| COX-2 | Forward 5′-TCT GCA GAG TTG GAA GCA CTC TA- 3′ | 55.6 | *Ittiudomrak et al. (2019)* |
| | Reverse 5′-GCC GAG GCT TTT CTA CCA GAA- 3′ | 57.2 | |
| GAPDH | Forward 5′-ACC CAC TCC TCC ACC TTT G- 3′ | 57.0 | *Ye et al. (2015)* |
| | Reverse 5′-ATC TTG TGC TCT TGC TGG G- 3′ | 55.3 | |

(Bioneer, Daejeon, South Korea) and specific oligonucleotide primers for the genes *IL-1β*, *TNF-α*, *iNOS* and *COX-2* (Table 1). All primers were synthesized by the Integrated DNA Technologies (IDT), Canada. Reactions were amplified and quantified in an Exicycler™ 96 Real-Time PCR System (Bioneer, Korea). Thermal cycles were completed at 95 °C for 2.5 min followed by 40 cycles at 95 °C for 30 s, 60 °C for 30 s, and 72 °C for 30 s (for *IL-1β*, *TNF-α* and *iNOS*) and 40 cycles at 95 °C for 30 s, 55 °C for 30 s, and 72 °C for 30 s (for *COX-2* and *gapdh*). The comparative threshold cycle (Ct) method was used to obtain relative quantities of mRNAs which were then normalized against *gapdh* as an endogenous control.

## Western blot analysis

A549 cells at $2 \times 10^6$ cells/well in 6-well plates (Nunc™) were pre-treated with various concentrations of the Pug-4 peptide (25–100 μM) for 1 h at 37 °C in 5% CO₂ and then infected with NTHi at a MOI of 10. After 9 h-incubation, supernatants were collected and cell pellets were washed with cold PBS pH 7.4 and lysed with RIPA lysis buffer (Amresco, OH, USA) containing 1 × Halt™ protease and phosphatase inhibitor cocktails (Pierce Biotechnology, Waltham, MA, USA). Proteins in the culture supernatant were precipitated using the methanol/chloroform method according to a previously described protocol (*Chanjitwiriya, Roytrakul & Kunthalert, 2020*). Nuclear and cytoplasmic proteins were extracted as reported previously (*Jantaruk et al., 2017*). The concentrations of extracted proteins were determined by using the Bradford protein assay kit (Bio-Rad, USA) according to the manufacturer's recommendations. Proteins extracted from cell lysates, cytoplasmic and nuclear fractions (25 μg), and supernatants (20 μL) were prepared in a Laemmli sample buffer (Bio-Rad, Hercules, CA, USA), loaded and separated using a 10% sodium dodecyl sulfate-polyacrylamide gel electrophoresis (SDS-PAGE), and subsequently transferred onto the nitrocellulose membranes (Bio-Rad) using a semi-dry transfer system (Bio-Rad). The immunoreactivity was performed as previously described (*Jantaruk et al., 2017*). The primary antibodies used were specific for IL-1β (3A6; Cell Signaling, USA), NLRP3 (D4D8T, Cell Signaling), caspase-1 (sc-56036; Santa Cruz Biotechnology, USA), iNOS (sc-7271; Santa Cruz Biotechnology), COX-2 (sc-1745; Santa Cruz Biotechnology), NF-κB p65

**Table 2** Minimal inhibitory concentration (MIC) and minimal bactericidal concentration (MBC) of Pug peptides against NTHi.

| Peptide | Concentration (µM) | |
| --- | --- | --- |
| | MIC[a] | MBC[a] |
| Pug-1 | >100 | >100 |
| Pug-2 | >100 | >100 |
| Pug-3 | >100 | >100 |
| Pug-4 | >100 | >100 |

Notes.
[a]The results represent the mean ± SD of two independent experiments performed in duplicate.

(sc-8008; Santa Cruz Biotechnology), and $\beta$-actin (ab170325; Abcam, USA). Anti-rabbit IgG, horseradish peroxidase (HRP)-linked antibody (7074S; Cell Signaling), or HRP-linked donkey anti-goat IgG (sc-2020; Santa Cruz Biotechnology), or peroxidase-conjugated AffiniPure goat anti-mouse IgG (115-035-003, Jackson ImmunoResearch Laboratories, USA) were used as secondary antibodies. The immunoreactive bands were developed using a Clarity™ Western ECL Substrate (Bio-Rad) and visualized using an ImageQuant LAS 4000 Biomolecular Imager (GE Healthcare Life Sciences, Chicago, IL, USA). The relative intensities of immunoreactive bands were analyzed using the ImageJ software and then normalized against the internal control $\beta$-actin.

## Statistical analysis

Data are expressed as mean ± standard deviation (SD) of independent experiments. Statistical analysis was performed using the SPSS version 23 software (SPSS, Chicago, IL, USA). A difference between the two groups was analyzed using a two-tailed student's $t$-test and a $p$-value less than 0.05 was considered statistically significant.

## RESULTS

### Effects of Pug peptides on growth of NTHi

This study used live NTHi in the infection experiments. We first assessed whether the Pug peptides (Pug-1, Pug-2, Pug-3 and Pug-4) had any effects on the viability of NTHi. NTHi susceptibility to Pug peptides was therefore examined. Exposure of NTHi to various Pug peptide concentrations ranging from 0.05 µM to 100 µM revealed no bacterial growth inhibition and because the MIC, and MBC values of all Pug peptides were >100 µM (Table 2). These results suggested that the Pug peptides at the concentrations studied had no impact on the viability of NTHi, and were thus used in the following experiments.

### Effects of Pug peptides on cell viability

The viability of RAW 264.7, differentiated THP-1, and A549 cells performed by an MTT assay was utilized to evaluate the toxicity of Pug peptides . As shown in Fig. 1, the viability of differentiated THP-1 or A549 cells upon exposure to Pug peptides at 25-100 µM is comparable and not statistically different from that of the untreated controls. In RAW 264.7 cells, a slight reduction of cell viability was observed after the exposure of Pug-1 at 100 µM (% viability, 88.52). Although statistically significant, this was not considered

cytotoxic. According to ISO 10993-5, percentages of cell viability above 80% are considered non-cytotoxic (*ISO 10993-5:2009 Biological Evaluation of Medical Devices. Part 5: Tests for In Vitro Cytotoxicity; International Organization for Standardization: Geneva 2009*). No significant changes in the viability of RAW 264.7 cells were found upon exposure to the other Pug peptides. No significant differences in cell viability between vehicle and untreated controls were observed in all three cell types. These results suggested that the Pug peptides at the studied concentrations as well as the vehicle did not induce significant cytotoxicity. Therefore, concentrations of up to 100 $\mu$M of the Pug peptides were considered a safe dose and could be used for the next studies.

## Effects of Pug peptides on NTHi-induced IL-1$\beta$ production

IL-1$\beta$ is a key mediator of airway inflammation in COPD (*Fu et al., 2015*; *Guo et al., 2022*; *Zou et al., 2017*). To evaluate the anti-inflammatory potential of Pug peptides, their effects on NTHi-induced production of the key pro-inflammatory cytokine IL-1$\beta$ were examined. As shown in Fig. 2, the levels of IL-1 $\beta$ in RAW264.7, differentiated THP-1, and A549 cells are markedly elevated upon infection with NTHi compared to the untreated controls. However, pre-treatment of Pug peptides at concentrations ranging from 2.5–100 $\mu$M resulted in a substantial reduction of NTHi-induced IL-1$\beta$ production. These effects were dose-dependent. Of particular note, this reduction was demonstrated in all studied cell types. Among the peptides tested, Pug-4 peptide exerted the most potent inhibitory activity with half-maximal inhibitory concentration ($IC_{50}$) values of 10.58, 2.11 and 10.93 $\mu$M for RAW 264.7, differentiated THP-1, and A549 cells, respectively. The inhibitory activity was less pronounced in Pug-2 peptide with $IC_{50}$ values of 19.84, 3.22 and 12.04 $\mu$M for RAW 264.7, differentiated THP-1, and A549 cells, respectively, followed by Pug-1 ($IC_{50}$ values of 28.26, 3.39 and 12.35 $\mu$M) and Pug-3 (28.37, 4.09 and 14.79 $\mu$M). Dexamethasone, one of the steroids used in COPD treatment, was also included in this study for efficacy comparison. Dexamethasone inhibited NTHi-induced production of IL-1$\beta$ in RAW 264.7 cells with an $IC_{50}$ value less than that of Pug-4 ($IC_{50}$ values of 6.89 and 10.58, respectively). Of significant note, however, the Pug-4 peptide exhibited greater inhibitory potency than dexamethasone in differentiated THP-1 cells ($IC_{50}$ values of 2.11 and 6.21 $\mu$M) while comparable potency was observed in A549 cells ($IC_{50}$ values of 10.92 and 10.08 $\mu$M). Based on these results, the Pug-4 peptide was selected for subsequent studies.

## Effects of Pug-4 peptide on NTHi-induced production of TNF-$\alpha$, NO and PGE$_2$

To further investigate anti-inflammatory activity of Pug-4 peptide, its effect on the NTHi-induced production of TNF-$\alpha$, NO, and PGE$_2$ in A549 cells was examined. These inflammatory mediators have also been associated with the pathogenesis of COPD (*Brown et al., 2023*; *Karadag et al., 2008*; *Xu et al., 2008*). The lung epithelial A549 cells were chosen in this experiment because airway epithelial cells are widely considered to be the first line of defense in the lungs and play a key role in organizing inflammatory and immune responses in the lungs (*Burgoyne, Fisher & Borthwick, 2021*; *De Rose et al., 2018*; *Gao et al., 2015*). This makes the lung epithelial cells a potential target for future therapeutic strategies

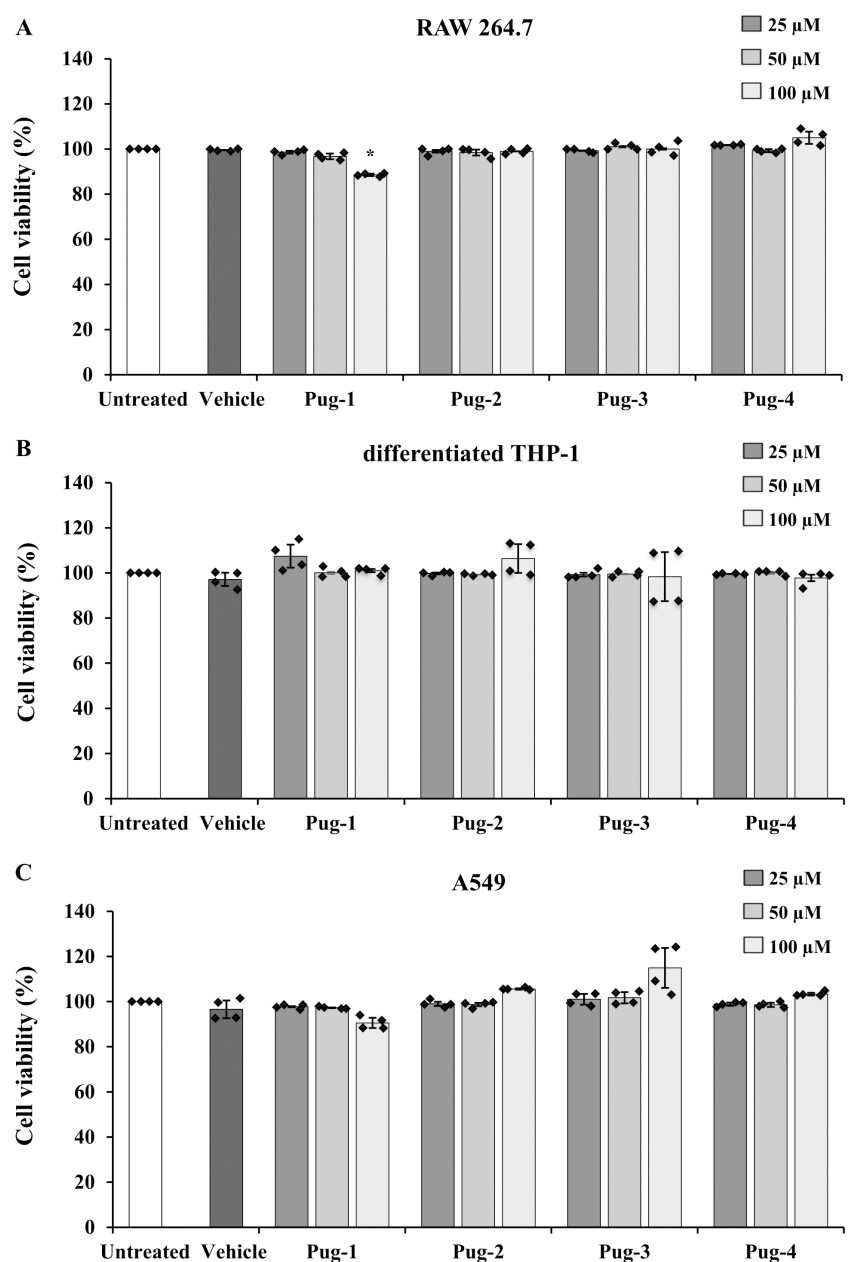

**Figure 1** **Pug peptides showed no cytotoxic effects in RAW 264.7, differentiated THP-1 and A549 cells.** RAW 264.7 (A), differentiated THP-1 (B) and A549 (C) cells were treated with 25, 50 and 100 μM of Pug peptides for 24 h, and cell viability was determined by an MTT assay. Cells without the test peptide served as an untreated control whereas DMSO-treated cells served as a vehicle control. Values are expressed as mean ± SD of two independent experiments performed in duplicate. An asterisk (*) indicates $p < 0.05$ compared with the untreated control.

aimed at dampening inflammatory responses (*Gao et al., 2015*; *Holtzman et al., 2014*). As presented in Fig. 3, the infection of A549 cells with NTHi results in the increased secretion of TNF-α, NO, and PGE₂, but significantly and dose-dependently decreases the secretion

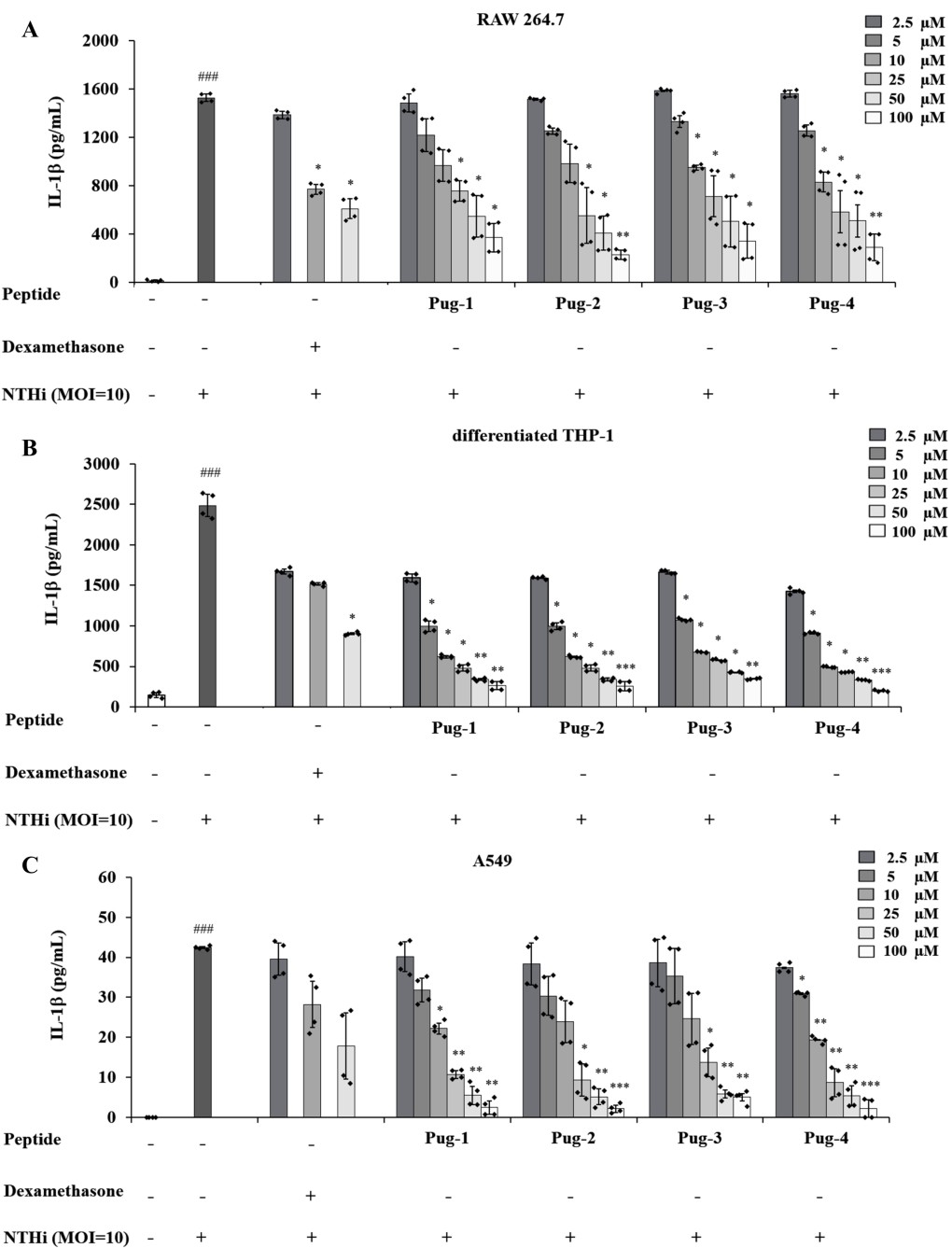

**Figure 2 Pug peptides inhibited the NTHi-induced IL-1β production.** RAW264.7 (A), differentiated THP-1 (B) and A549 (C) cells were treated with different concentrations of Pug peptides or dexamethasone for 1 h, then infected with NTHi at a MOI of 10. The levels of IL-1β in culture supernatant were determined by sandwich ELISA. Cells without the test peptide served as an untreated control whereas DMSO-treated cells served as a vehicle control. Values are expressed as mean ± SD of two independent experiments performed in duplicate. ###, $p < 0.001$ compared with untreated control cells; *, $p < 0.05$; **, $p < 0.01$ and ***, $p < 0.001$ compared with the NTHi-infected cells.

of TNF-$\alpha$, NO, and PGE$_2$ when treated with Pug-4. The Pug-4 peptide strongly inhibited NTHi-induced production of TNF-$\alpha$, NO, and PGE$_2$ with % inhibition ranging from 48.60–100.00, 83.50–94.33 and 49.42–80.97%, respectively. It is significant to note that Pug-4 at 100 $\mu$M completely inhibited the production of TNF-$\alpha$ in NTHi-induced A549 cells and suppressed the production of NO and PGE$_2$ to levels close to that of unstimulated A549 cells ($p > 0.05$).

### Effects of Pug-4 peptide on IL-1$\beta$, TNF-$\alpha$, iNOS and COX-2 mRNA expression in NTHi-infected A549 cells

The inflammatory mediators are regulated primarily at the mRNA level. To understand whether Pug-4 affected the activation of gene encoding for IL-1$\beta$, TNF-$\alpha$, inducible nitric oxide synthase (iNOS), and cyclooxygenase (COX)-2 in NTHi-infected A549 cells, qPCR was performed. As shown in Fig. 4, the expression of IL-1$\beta$, TNF-$\alpha$, iNOS, and COX-2 mRNA is significantly elevated in NTHi-infected A549 cells and dramatically reduced with increasing concentrations of Pug-4. These results showed that Pug-4 suppressed the expression of these inflammatory mediator genes at the transcriptional level.

### Effects of Pug-4 peptide on iNOS and COX-2 protein expression in NTHi-infected A549 cells

COX-2 is the key enzyme required for the conversion of arachidonic acid to PGE$_2$ while iNOS is the essential enzyme required for the production of NO from L-arginine. We next examined the effects of Pug-4 peptide on the expression of iNOS and COX-2 proteins. This was carried out by Western blotting. As demonstrated in Fig. 5, expression levels of iNOS and COX-2 proteins in A549 cells upregulate after NTHi infection, while the Pug-4 peptide significantly suppresses NTHi-induced iNOS and COX-2 protein expression in a dose-dependent manner.

### Effects of Pug-4 peptide on NTHi-induced activation of NF-$\kappa$B signaling pathway in A549 cells

We further investigated whether Pug-4 exerted a regulatory role on the NTHi-induced NF-$\kappa$B activation signaling pathway in A549 cells for two reasons: first, the nuclear factor-kappa B (NF-$\kappa$B) pathway is essential in inflammatory responses to various pathogen infections, including NTHi (*Shuto et al., 2001*); second, the translocation of the transcription factor NF-$\kappa$B from cytosol to the nucleus is critically required for the activation of inflammatory gene transcription. Protein expression of NF-$\kappa$B p65, the major component of NF-$\kappa$B activation in the cytosolic and nuclear fractions, was examined. The results from Western blotting in Fig. 6 show that NF-$\kappa$B p65 is less expressed in the cytosol and strongly expressed in the nuclear fraction after infection with NTHi. The results suggest that NTHi infection triggers the translocation of the NF- $\kappa$B p65 subunit to the nucleus, thereby activating the NF- $\kappa$B pathway. Treatment with Pug-4 significantly reduced the expression of nuclear NF-$\kappa$B p65 protein in a concentration-dependent manner, suggesting that Pug-4 suppressed the translocation of NF-$\kappa$B p65 from the cytosol to the nucleus. The suppression of nuclear NF-$\kappa$B p65 protein expression to a level close to that of the untreated cells was observed with the treatment of Pug-4 at 100 $\mu$M ($p > 0.05$).

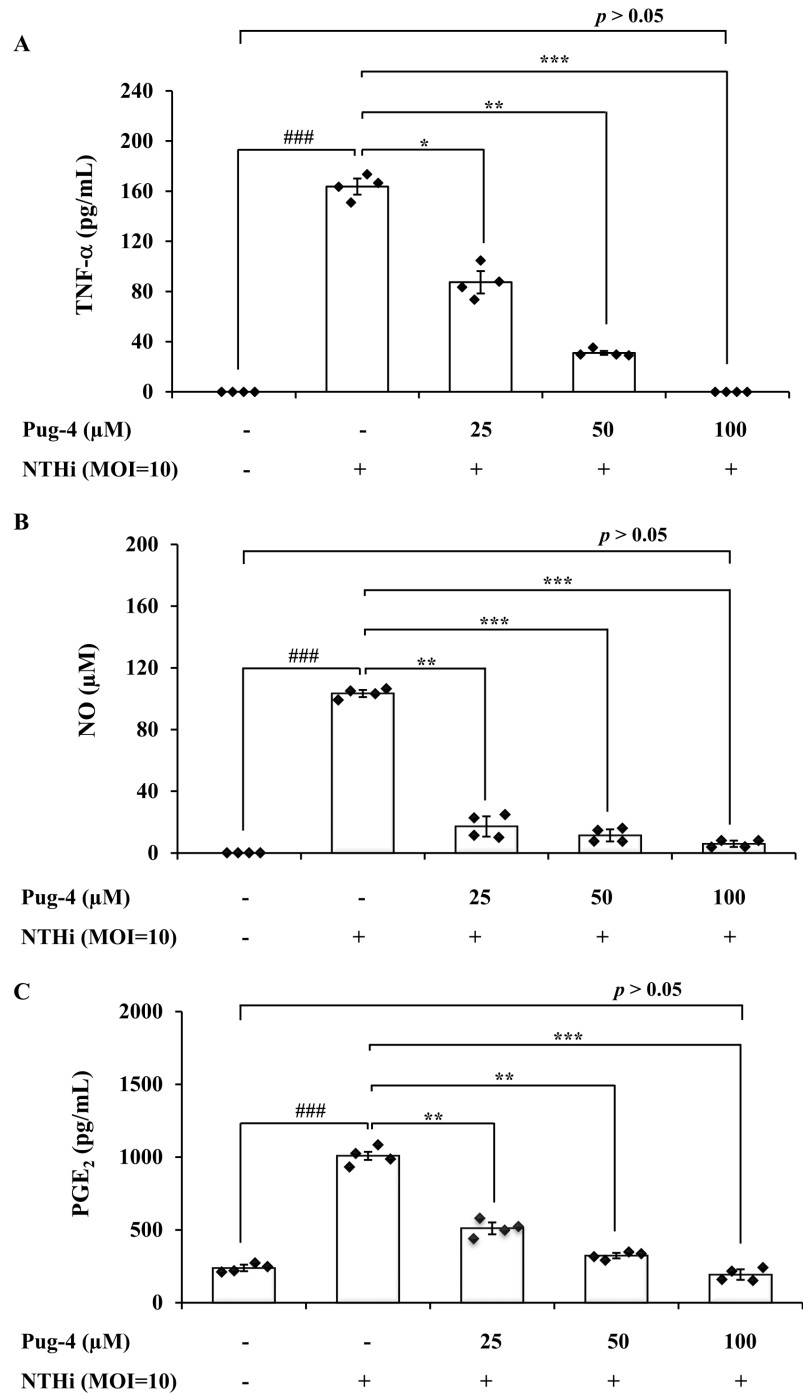

**Figure 3** **Pug-4 peptide inhibited the NTHi-induced production of TNF-a, NO and PGE$_2$.** A549 cells were treated with different concentrations of Pug-4 peptide for 1 h prior to the infection of NTHi at a MOI of 10. After 9 h-incubation, culture supernatants were collected and analyzed for TNF-a (A), NO (B) and PGE$_2$ (C) by sandwich ELISA, Griess assay and enzyme-immunoassay, respectively. Values are expressed as mean ± SD of two independent experiments assayed in duplicate. ###, $p < 0.001$ compared with untreated A549 cells. *, $p < 0.05$; **, $p < 0.01$ and ***, $p < 0.001$ compared with the NTHi-infected A549 cells.

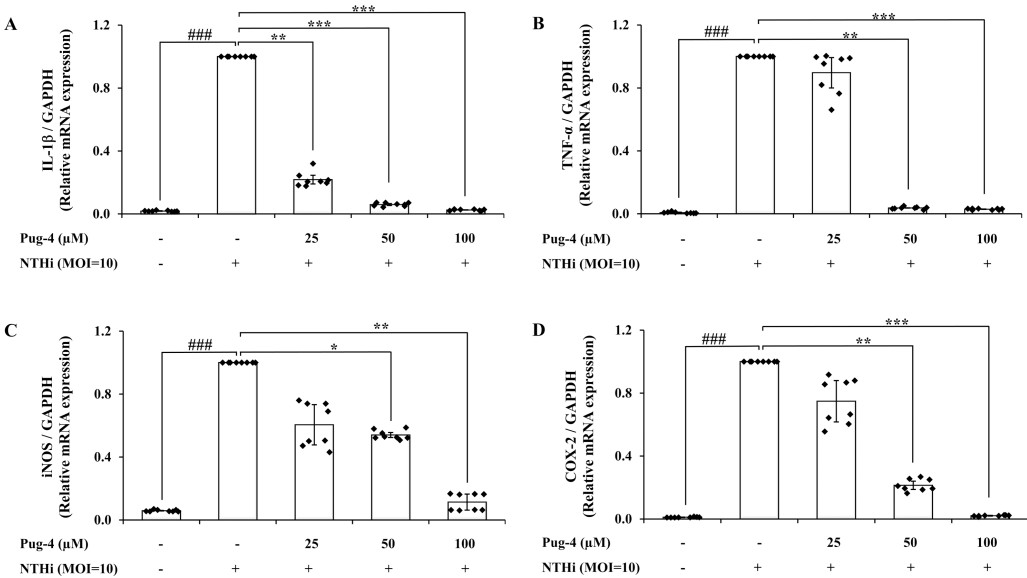

**Figure 4** **Pug-4 peptide inhibited IL-1$\beta$, TNF-$\alpha$, iNOS and COX-2 mRNA expression in NTHi-infected A549 cells.** A549 cells were treated with Pug-4 peptide for 1 h prior to the infection with NTHi at a MOI of 10 for 9 h. The levels of IL-1$\beta$ (A), TNF-$\alpha$ (B), iNOS (C) and COX-2 (D) mRNA expression were determined by qPCR. Values are expressed as mean ± SD of two independent experiments performed in quadruplicate. ###, $p < 0.001$ compared with untreated A549 cells; *, $p < 0.05$; **, $p < 0.01$ and ***, $p < 0.001$ compared with the NTHi-infected A549 cells.

## Effects of Pug-4 peptide on NTHi-induced activation of NLRP3 inflammasome pathway in A549 cells

Previous studies demonstrated that NTHi induced NLRP3 inflammasome activation and led to caspase-1-dependent secretion of IL-1$\beta$ (*Rotta Detto Loria et al., 2013*). To access the effects of Pug-4 on NLRP3 inflammasome activation in NTHi-infected A549 cells, Western blotting to detect inflammasome key components was performed. As shown in Fig. 7, significant increases in the expression of pro-caspase-1, intermediate caspase-1, and caspase-1 are evident in NTHi-infected A549 cells, which were dose-dependently suppressed with the treatment of Pug-4 (Figs. 7A and 7B). Similarly, marked increases in the expression of pro-IL-1$\beta$ and IL-1$\beta$ were observed in NTHi-infected A549 cells, which were also significantly suppressed by increasing the concentrations of Pug-4 (Figs. 7A and 7C). There was a significant decrease in the NLRP3 protein expression with Pug-4 peptide treatment, especially at concentrations of 100 µM (Figs. 7A and 7D). These results showed that Pug-4 decreased the expression of core components involved in NLRP3 inflammasome activation.

## DISCUSSION

Given that NTHi infection is an important factor in COPD, largely associated with chronic lung inflammation and risk of exacerbation, and currently lacking effective medication for treatment, developing novel therapeutics targeting excessive inflammation is of significant focus. We explored the regulatory potential of four pomegranate-derived peptides on

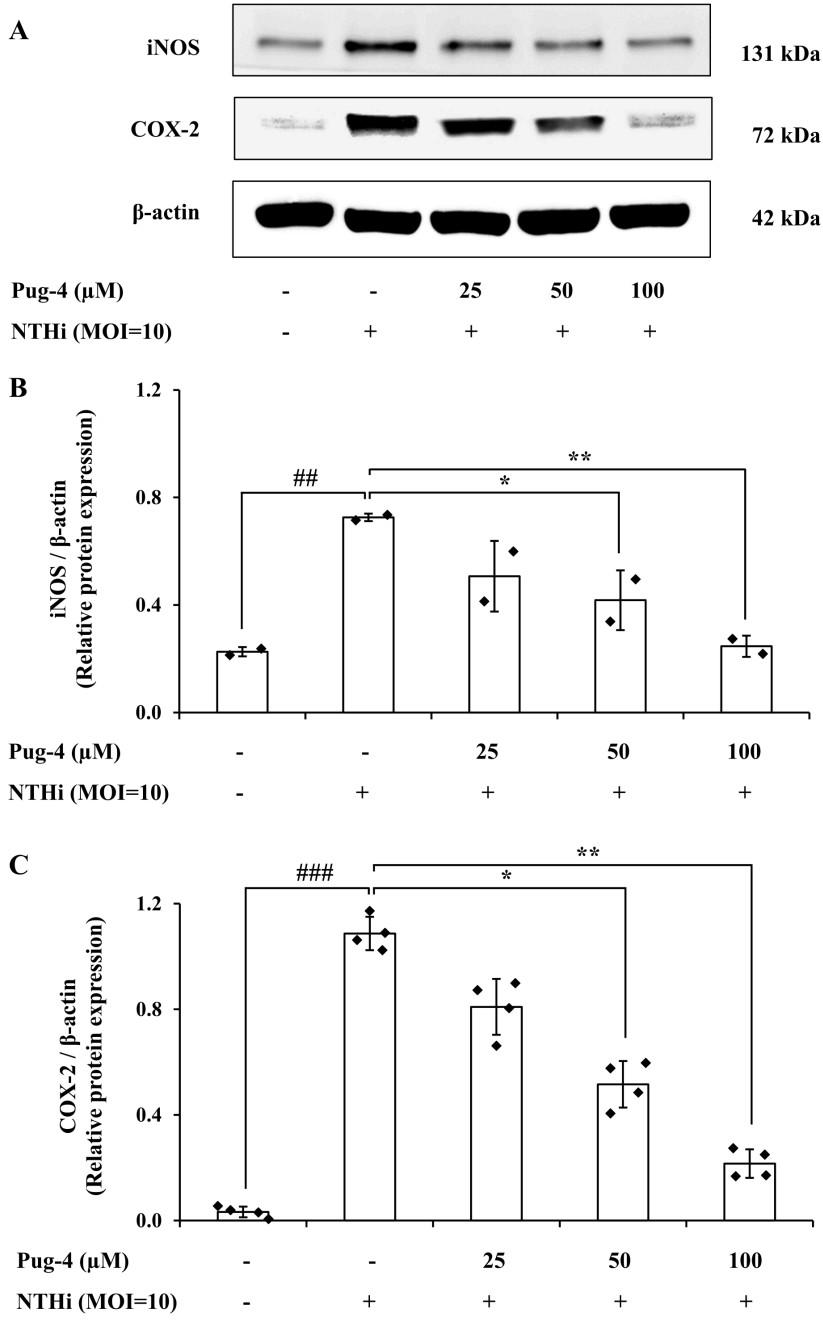

**Figure 5 Pug-4 peptide inhibited iNOS and COX-2 protein expression in NTHi-infected A549 cells.**
A549 cells were treated with Pug-4 peptide for 1 h prior to the infection with NTHi at a MOI of 10 for 9 h. The levels of iNOS and COX-2 protein expression were examined by Western blotting (A). Bar diagrams showing densitometric analysis of the relative protein expression of iNOS/$\beta$-actin (B) and COX-2/ $\beta$-actin (C), quantified using ImageJ software. Values are expressed as mean ± SD of two independent experiments. ##, $p < 0.01$ and ###, $p < 0.001$ compared with untreated A549 cells; *, $p < 0.05$ and ** $p < 0.01$ compared with the NTHi-infected A549 cells.

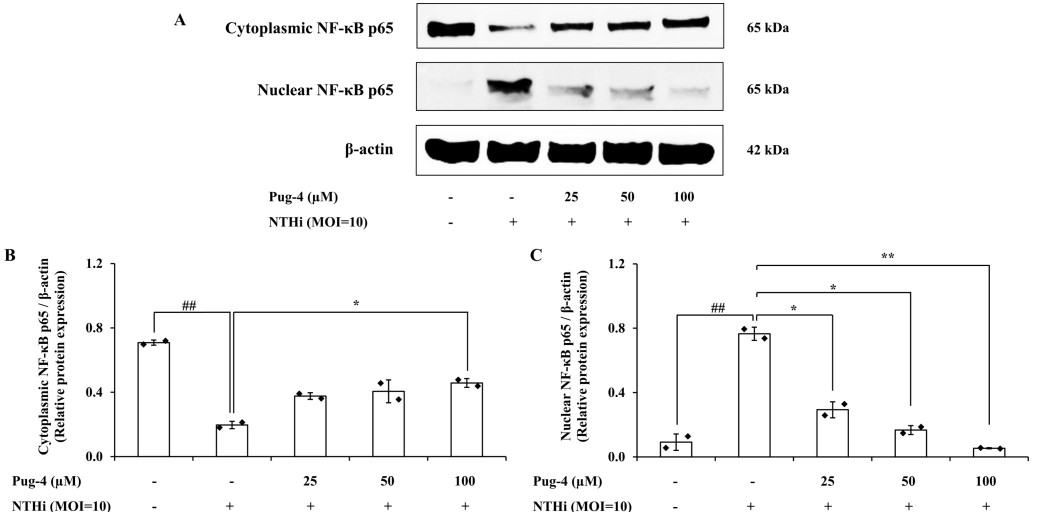

**Figure 6** **Pug-4 peptide inhibited NTHi-induced NF-$\kappa$B activation in A549 cells.** A549 cells were treated with Pug-4 peptide for 1 h prior to the infection with NTHi at a MOI of 10. After 30 min, the expression of cytosolic and nuclear protein fractions of NF-$\kappa$B p65 protein was determined by Western blotting (A). Bar diagrams showing densitometric analysis of the relative expression of cytoplasmic NF-$\kappa$B p65/$\beta$-actin (B) and nuclear NF-$\kappa$B p65/$\beta$-actin (C), quantified using ImageJ software. Values are expressed as mean ±SD of two independent experiments. ##, $p < 0.01$ compared with untreated A549 cells; *, $p < 0.05$ and ** $p < 0.01$ compared with the NTHi-infected A549 cells.

NTHi-induced inflammation. Live NTHi was used to mimic an actual infection and their inhibitory activities against the production of IL-1$\beta$, a pivotal mediator in COPD, were evaluated. The initial focus is on the proinflammatory cytokine IL-1$\beta$ because it plays a significant role in initiating and maintaining airway inflammation (*Fu et al., 2015*; *Guo et al., 2022*; *Zou et al., 2017*) and increased IL-1$\beta$ secretion has been reported in both stable and exacerbated COPD (*Bafadhel et al., 2011*; *Pauwels et al., 2011*). The results in this study showed that the Pug peptides to varying degrees inhibited the production of IL-1$\beta$ induced by NTHi and showed that the effects were not due to direct cytotoxicity. When considering the inhibition degree values among the peptides tested, Pug-4 appeared to be the most effective; the lowest IC$_{50}$ toward NTHi-induced IL-1$\beta$ production was clearly seen with the treatment of Pug-4. Interestingly, the potent inhibitory effect of Pug-4 was demonstrated in all studied cell types; in murine (RAW264.7) and human(differentiated THP-1) macrophages, as well as human lung epithelial cells (A549), suggesting that the capability of Pug-4 peptide to inhibit NTHi-induced IL-1$\beta$ production in diverse innate immune cells is high. Its inhibitory potency was further supported by the observations that the IC$_{50}$ values of Pug-4 against NTHi-induced IL-1$\beta$ secretion were comparable and even less than that of the reference drug dexamethasone in human lung epithelial cells (A549) and THP-1 macrophages, respectively. Besides IL-1$\beta$, other inflammatory cytokines including TNF-$\alpha$, NO, and PGE$_2$ have been implicated in the pathogenesis of COPD (*Brown et al., 2023*; *Karadag et al., 2008*; *Xu et al., 2008*). Elevated levels of sputum PGE$_2$ have also been reported to be associated with increased respiratory symptoms and a

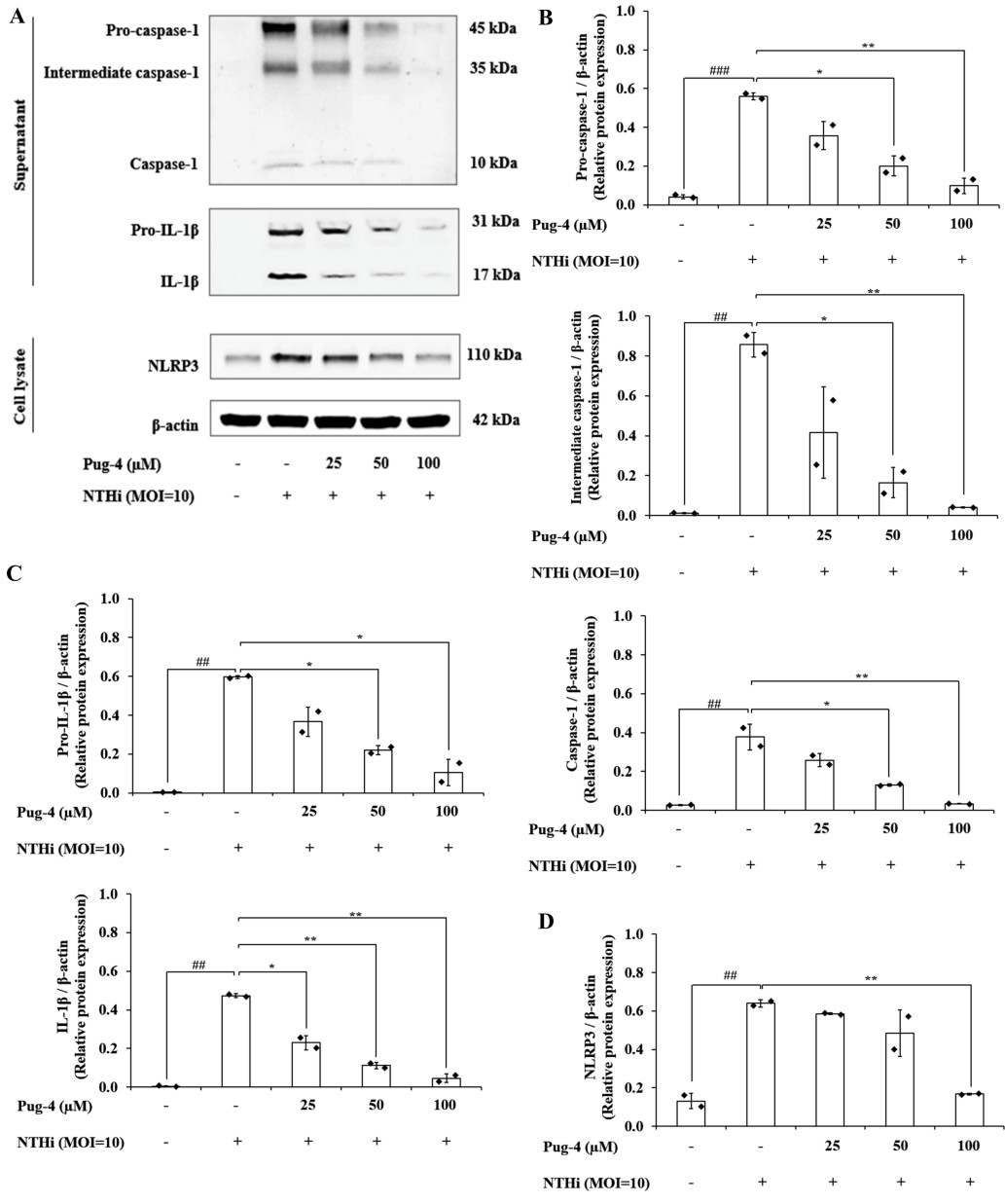

**Figure 7** **Pug-4 peptide suppressed NTHi-induced activation of NLRP3 inflammasome pathway in A549 cells.** A549 cells were treated with Pug-4 peptide for 1 h prior to the infection with NTHi at a MOI of 10 for 9 h. Protein expressions from supernatants and cell lysates were examined by Western blotting. The Western blot images of protein bands are the representative of separate experiments (A). Bar diagrams showing densitometric analysis of the relative expression of pro-caspase-1/$\beta$-actin, intermediate caspase-1/ $\beta$-actin, caspase-1/$\beta$-actin (B), pro-IL-1$\beta$/ $\beta$-actin, IL-1$\beta$/$\beta$-actin (C) and NLRP3/ $\beta$-actin (D), quantified using ImageJ software. Data are expressed as mean ±SD of two independent experiments. ##, $p < 0.01$ and ###, $p < 0.001$ compared with untreated A549 cells; *, $p < 0.05$ and ** $p < 0.01$ compared with NTHi-infected A549 cells.

history of exacerbation among individuals with COPD (*Tejwani et al., 2023*). The results obtained from this study also showed that the Pug-4 peptide strongly reduced TNF-$\alpha$, NO, and PGE$_2$ levels elicited from A549 cells infected with NTHi. This observation additionally strengthens the evidence of the powerful inhibitory activity of Pug-4. The findings herein clearly demonstrated the effectiveness of Pug-4 against NTHi-induced inflammation.

During NTHi infection, the NLRP3 inflammasome in respiratory cells and tissues is upregulated and leads to caspase-1-dependent secretion of IL-1$\beta$ (*Chen et al., 2018*; *Rotta Detto Loria et al., 2013*). Activation of NLRP3 inflammasome employs a two-step mechanism. The first step involves the priming signal which is induced by the toll-like receptor (TLR)/ NF-$\kappa$B pathway in order to upregulate the expression of NLRP3 and pro-IL-1$\beta$. The second step is the oligomerization of NLRP3 and the assembly of the multi-protein complex NLRP3, the adaptor protein ASC, and pro-caspase-1 to trigger the autoproteolysis of caspase-1 and the cleavage of pro-IL-1$\beta$ (31 kDa) into a mature form (17 kDa) (*Akbal et al., 2022*; *Guarda & So, 2010*; *Guo, Callaway & Ting, 2015*). Since NLRP3 inflammasome activation has been implicated in bacterial-driven exacerbation in COPD (*Rotta Detto Loria et al., 2013*), the involvement of Pug-4 in both of the prerequisite steps of inflammasome activation was investigated. We found that Pug-4 strongly inhibited the expression of NF-$\kappa$B p65 protein in the nucleus (Fig. 6). This suggests that Pug-4 inhibits the translocation of NF-$\kappa$B p65 to the nucleus and thus indicates the inhibition of NF-$\kappa$B activation (*Yu et al., 2020*). The inhibition of the NF-$\kappa$B pathway was also observed by a decrease in the transcription of the NF-$\kappa$B target genes, determined by the expression of IL-1$\beta$, TNF-$\alpha$, iNOS, and COX-2 mRNA (Fig. 4), which correlated with those of protein expression (Fig. 5). The iNOS and COX-2 are the catalytic enzymes required for the production of NO and PEG$_2$. Our results suggested that Pug-4 inhibited the NO and PEG$_2$ production by down-regulating iNOS and COX-2 at both the mRNA and protein levels. We also observed that Pug-4 suppressed the expression of NLRP3 and pro-IL-1$\beta$ proteins (Fig. 7). Furthermore, the inhibition of pro-caspase-1 and caspase-1 protein expression was clearly seen (Fig. 7), suggesting that Pug-4 suppressed the NTHi-mediated cleavage of caspase-1. Consistent with these observations, mature IL-1 $\beta$ production was remarkably decreased upon the treatment of Pug-4 (Fig. 7). These results indicated that Pug-4 inhibited both critical checkpoints of inflammasome activation. Collectively, our results clearly demonstrated that Pug-4 inhibited NTHi-induced inflammation through the NF-$\kappa$B and NLRP3 inflammasome pathways. In this study, Pug-4 was added to lung epithelial cells prior to NTHi infection and the inhibition of key inflammatory mediator production was observed. Some concerns remain regarding whether the Pug-4 binds to a specific receptor or directly penetrates the lung epithelial cells and later interferes with essential signaling molecules involved in the production of inflammatory mediators. This cannot be ascertained from our experiment parameters and additional studies are certainly needed to clarify such an issue. Nevertheless, our findings provided clear evidence of the ability of Pug-4 to strongly inhibit NF-$\kappa$B and NLRP3 inflammasome pathways and subsequently inhibit the production of key inflammatory mediators induced by NTHi infection. Regulation of the NF-$\kappa$B and NLRP3 inflammasome offers promising pharmaceutical options for treatment of inflammatory diseases including those induced

by bacterial infections (*Chernikov et al., 2021*; *Leszczyńska, Jakubczyk & Górska, 2022*; *Swanson, Deng & Ting, 2019*; *Zhang et al., 2021*). Our findings therefore firmly indicated the potential of the Pug-4 peptide to regulate NTHi-mediated inflammation.

## CONCLUSION

In summary, this study demonstrated for the first time that a pomegranate-derived peptide, the Pug-4 peptide, exerted potent anti-inflammatory activity against NTHi-induced inflammation. Pug-4 strongly suppressed the key pro-inflammatory mediator IL-1$\beta$, TNF-$\alpha$, NO, and PGE$_2$ and inhibited the NF-$\kappa$B pathway and NLRP3 inflammasome activation in lung epithelial cells when infected with NTHi. Pug-4 therefore shows promise as a novel therapeutic agent for the treatment of NTHi-mediated inflammation and other respiratory inflammatory diseases.

### Funding
This work was supported by the Royal Golden Jubilee Ph.D. (RGJ-PHD) Program Scholarship (Grant No. PHD/0050/2559). The funders had no role in study design, data collection and analysis, decision to publish, or preparation of the manuscript.

### Grant Disclosures
The following grant information was disclosed by the authors:
Royal Golden Jubilee Ph.D. (RGJ-PHD) Program Scholarship: PHD/0050/2559.

### Competing Interests
The authors declare there are no competing interests.

### Author Contributions

- Pornpimon Jantaruk conceived and designed the experiments, performed the experiments, analyzed the data, prepared figures and/or tables, authored or reviewed drafts of the article, and approved the final draft.
- Sittiruk Roytrakul conceived and designed the experiments, authored or reviewed drafts of the article, and approved the final draft.
- Anchalee Sistayanarain analyzed the data, authored or reviewed drafts of the article, and approved the final draft.
- Duangkamol Kunthalert conceived and designed the experiments, analyzed the data, prepared figures and/or tables, authored or reviewed drafts of the article, and approved the final draft.

### Data Availability
The raw measurements are available in the Supplementary Files.

## Supplemental Information

Supplemental information for this article can be found online at http://dx.doi.org/10.7717/peerj.16938#supplemental-information.

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
