# Peer review of "The pomegranate-derived peptide Pug-4 alleviates nontypeable Haemophilus influenzae-induced inflammation by suppressing NF-kB signaling and NLRP3 inflammasome activation"

_PeerJ, doi:10.7717/peerj.16938_

## Round 0.1 · original submission · Major Revisions

Kindly submit a thoroughly revised version and a detailed rebuttal letter addressing all the concerns raised by the reviewers.

Reviewer 1 ·

Basic reporting

Authors claim in the title that pug-4 alleviates NTH-induced inflammation by suppressing NFKB and NLRP3 inflammasome pathways. However, both of them are 2 different pathways. They should re-phrase their title.

Experimental design

1. This manuscript includes results with fundamental misrepresentation of protein expression data, namely, in figure 7 procaspase expression (a constitutively expressed protein) is lowering with PUG-4 treatment (not detectable in 5th lane) while having equal b-actin levels. This cannot be true. Similarly, if caspase-1 expression is down then cleaved caspase-1 will also be lower and cannot cleave pro-IL-1 beta so claiming NLRP3 specific effect is certainly an over-exaggeration of data.
2.Effects of Pug peptides on growth of NTHi: Figure is missing for this result.
3. Please provide details explanation why were these cell lines used for this study and one sentence about these cell lines which cell type, mouse/human cell line is used, etc.,? Also, please provide individual data points. Also, give no. of experiments done. Explain in figure legend difference between untreated control or vehicle?
4. Please explain in the results why was dexamethasone used in this experiment? The lowest concentration 2.5uM and 5uM does not reduce NTHi-induced IL-1b production in Raw cell lines(Fig2A), please re-write results and mention at which dose it significantly reduces il-1b production.
5. Please provide an explanation why authors used only A549 epithelial cells for NO,TNFa and PGE2? Please mention in the figure legends which method was used to analyze NO,TNFa and PGE2.
6. Explain what is COX-2 and iNOS and why was it measured?
7.Figure 4 and Figure 5 have no labels like A,B,C,D.
8.To confirm the role of NF-Kb, inhibitor of NFKb should be used as a control.

Validity of the findings

This manuscript includes results with fundamental misrepresentation of protein expression data, in figure7. Therefore, the manuscript results do not validate the claimed conclusions.

Additional comments

1. Please provide individual data points for each graph of the results.
2. Authors should explain in each result why the experiment was performed and provide some background with every result. It is easier to understand for a wider audience.
3. Results need to be explained in detail.
4. Why was MOI 10 NTHi used in the experiments? Did they do a kinetics study?
5. Since Figure 4 and Figure 5 has same conclusions but different techniques, it should be combined in the same results and figures.
6. Conclusions with each results are also missing.
7. The English grammar should be improved for the whole manuscript. I suggest you have a colleague who is proficient in English and familiar with the subject matter review your manuscript.

·

Basic reporting

.

Experimental design

.

Validity of the findings

.

Additional comments

In this manuscript, Jantaruk et al, demonstrate the anti-inflammatory effects of a pomegranate-derived peptide, Pug-4 against NTHi-induced inflammation. The authors have provided substantial evidence to show that Pug-4 peptide strongly inhibits NF-κB and NLRP3 inflammasome pathways and the production of key inflammatory mediators induced by NTHi infection. However, there are certain concerns that the authors need to address for the manuscript to be considered for publication by PeerJ. These include -

1. The introduction is a bit lengthy. Shorten it.
2. “Uniquely, the synthesis and secretion of mature IL-1β is under tight transcriptional and post-transcriptional control required the participation of nucleotide-binding oligomerization domain-like receptor containing pyrin domain 3 (NLRP3) inflammasome” – rephrase the sentence and add reference.
3. Expand all the abbreviations used in the manuscript (examples – BAL etc.)
4. It will be good to end the introduction with a sentence stating the implications/big picture of this study.
5. Results for the section, “Effects of Pug peptides on growth of NTHi” is missing. This data should be included.
6. In Figure 2, data showing the half-maximal inhibitory concentration (IC50) values of Pug 4 for RAW 264.7, differentiated THP-1 and A549 cells should be included.
7. It should be explained why A549 cells were selected for further experiments.
8. “It is significant to note that Pug-4 peptide at 100 μM completely inhibited the production of TNF- in NTHi-induced A549 cells and suppressed the production of NO and PGE2 to the level close to (p > 0.05) that of unstimulated A549 cells”. Data should be included to show that the cells are viable when treated with 100 μM Pug-4 and NTHi, and that the reduction in TNF- levels were not because the cells were not viable.
9. It should be stated either in the figure legends/methods, how many times each experiment was performed.
10. It would be beneficial for the readers if the authors include a summary statement explaining the results (and what the results imply/suggest) at the end of each result section.
11. What statistics was used for Figure 1? It seems like 100 μM Pug-1 decreases cell viability in RAW 264.7 and A549 cells. State this in the text.
12. It seems Pug-2/3 had similar effect as Pug-4. It should be described in detail why this peptide was chosen and not the others.
13. Figure 3-6 include the figure labels and incorporate them in the figure legends.
14. Figure 4 – Statistics is missing for 25 μM TNF-a
15. Figure 6 – Explain the physiological significance of altered NF-κB in the nuclear and cytosolic fraction.
16. Figure 7 – Some of the statistics is missing.
Minor comments-

There are certain typos and some grammatically incorrect statements. Please read thoroughly and fix them.

---

## Round 0.2 · accepted · Accept

The authors have addressed all the reviewer comments and it is now accepted in PeerJ.

·

Basic reporting

The authors have addressed all my comments. The manuscript is now acceptable for publication.

Experimental design

N/A

Validity of the findings

N/A

Additional comments

N/A